# Dynamic Analysis of Fuzzy Systems

**Manuel Barraza** [1,2,*] , **Fernando Matía** [1,3] **and Basil Mohammed Al-Hadithi** [1,4]

1. Centre for Automation and Robotics (UPM-CSIC), Universidad Politécnica de Madrid, 28006 Madrid, Spain
2. Departamento de Ingeniería Eléctrica-Electrónica, Universidad de Tarapacá, Arica 1010069, Chile
3. Department of Automatic Control, Electrical and Electronic Engineering and Industrial Computing, Universidad Politécnica de Madrid, 28006 Madrid, Spain
4. Department of Electrical, Electronics, Control Engineering and Applied Physics, Higher Technical School of Industrial Design and Engineering, Universidad Politécnica de Madrid, 28012 Madrid, Spain
* Correspondence: mabarrazar@academicos.uta.cl

**Abstract:** In this work, a new methodology for the dynamic analysis of non-linear systems is developed by applying the Mamdani fuzzy model. With this model, parameters such as settling time, peak time and overshoot will be obtained. The dynamic analysis of non-linear fuzzy systems with triangular membership functions is performed, and linguistic variables describing overly complex or ill-defined phenomena are used to fit the model. Scaling factors will simplify the modification of the variables, making them easier to find the system model. The specifications of second-order characteristics in the time domain, such as overshoot and peak time, will be represented graphically. As a case study, the proposed methods are implemented to analyse the dynamics of a tank and a simple pendulum for first-order and second-order systems, respectively, where it is observed that the proposed methodology offers highly positive results.

**Keywords:** fuzzy systems; dynamic analysis; modelling; settling time; peak time; overshoot; scaling factors; non-linear systems

## 1. Introduction

The most common models in fuzzy control, the Mamdani model [1] and Takagi–Sugeno (T-S) model [2], are traditionally used to model non-linear systems. The main difference lies in the consequent part of the fuzzy rules. Mamdani fuzzy systems use fuzzy sets as the consequents of the rules, while T-S fuzzy systems use affine functions of the input variables as the consequents of the rules. From a linguistic perspective, Mamdani-type fuzzy systems are understandable since fuzzy linguistic variables are used in both premises and consequents [3]. On the other hand, in the T-S model, the number of free parameters is restricted by a suitable structure of the consequent functions, and the antecedent of each rule defines a validity region for the corresponding affine consequent model. It can be considered an optimization problem [4,5]. This work deals with the dynamic analysis of non-linear systems rather than control design. Therefore, the Mamdani model would be the appropriate choice.

The structure of the fuzzy model using the product-sum-gravity method was used [6,7] to obtain a flat surface in the inference map and then modify the membership functions with linguistic operators. The min-max-gravity method [8] does not obtain a flat surface in the inference map due to the non-linearity of these operators.

Knowledge of a system is essential for applying fuzzy logic. Combining it with linguistic information (by experts) and numerical data improves the ability to represent complex non-linear functions with simple local linear models [9]. Although these linguistic descriptions are not precise, they provide essential information about the system. A human operator can determine a set of successful control rules based solely on the linguistic descriptions of the process. This is based on Mamdani Fuzzy Inference Systems in linguistic

synthesis [10]. In recent work, in [11], the authors develop the "Linguistic Compositional Variable" to represent mole fractions of components of each phase as a fuzzy variable. As a result, the number of rules is significantly reduced. In [12], the authors use the cloud model to represent the linguistic variables. Instead, [13] uses a Mamdani Fuzzy Inference Network (MFIN) with an advanced optimization technique. The study of evaluative linguistic expressions has crossed the field of theoretical linguistics and has attracted interest in very different research areas, such as artificial intelligence, psychology, or cognitive linguistics [14].

Applying a simple arithmetic operation on the fuzzy sets will modify them. It tends to consider the adverb "very" interpretation as the result of squaring a membership function, which causes the membership function to take smaller values, concentrating on the larger ones. Zadeh [15] called this concentration. The opposite effect is dilation, which would correspond to the interpretation of the adverb "more or less", which would be implemented as the square root of the membership function. These are linguistic variables used to describe overly complex or ill-defined phenomena typical of non-linear systems. These modifiers have been used for interpretable context adaptation [16,17]. In [18], an extended fuzzy hedge set is used to create a more detailed classification of client sentiments, achieving better performance than without them. In some approaches, they are used to create more compact rules, such as [19].

The basic first-order system is described with a single real root, while the underdamped second-order system has a single pair of complex roots. Many devices and processes are practically well-modelled by either of these two classes, and complex systems can be considered a combination of several first-order and/or second-order forms [20].

In previous work, the authors have studied the transient response of feedback systems [21], using the Takagi–Sugeno fuzzy model for system identification [22–24] or applying fuzzy logic to improve Kalman filtering [25,26]. In this work, the analysis of the transient response of fuzzy systems is proposed using the Mamdani model, a new approach that has not been previously performed.

The literature on fuzzy model identification has not adequately addressed issues such as efficient experiment design, model class, modelling error, target control specifications, etc. Very few publications address the dynamic properties of fuzzy models [27]. Today, the dynamic analysis of fuzzy systems tends to demonstrate system stability, which is an important problem for controller designs [28–31]. To guarantee stability, a mathematical model of the plant is needed. There are many works on fuzzy control analysis using phase plane methods [32–34]. However, these works focus on control and not on dynamic analysis.

This work proposes an analysis of the dynamical behaviour of first and second-order fuzzy systems and to be able to predict or approximate parameter values, such as their settling time and overshoot. To predict the dynamical behaviour of a first or second-order system, we will propose the following steps: (1) use triangular membership functions to identify the system, and then fit the model by applying the concentration or dilation linguistic operator or using known membership functions to fit the model; (2) calculate the corresponding time characteristics with the obtained results; (3) estimate how much the real dynamic characteristics deviate from the previous ones through the plots we propose experimentally.

The main aim of this work is to provide a dynamic analysis of fuzzy systems. The paper is organized as follows: Section provides a review of fuzzy systems for dynamic modelling. The Fuzzy Mamdani-type model is explained in Section 2. The settling time for first-order fuzzy systems is calculated in Section 3. The performance of fuzzy second-order dynamical systems is presented in Section 4. In Section 5, the proposed methods are applied to tank and simple pendulum systems.

## 2. Mamdani Model

In a general case, the dynamics of the non-linear system are described by the following equation:

$$x_{n+1} = f(\mathbf{x}) \tag{1}$$

being $\mathbf{x} = [x_1...x_n]^T$ and $f$ a continuous or discrete non-linear function [35]. The main aim is to build non-linear models capable of reflecting the dynamics of $f$. System modelling helps to visualise the behaviour and basic structure of processes. Let us suppose that $X_l^{i_l}$ are the fuzzy sets for input $x_l$, $\forall i_l = \{1,...,r_l\}$, $\forall l = \{1,...,n\}$. $r_l$ is the number of fuzzy sets for $x_l$, and $\mu_{X_l^{(i_l)}}(x_l)$ are the corresponding membership functions; see Figure 1.

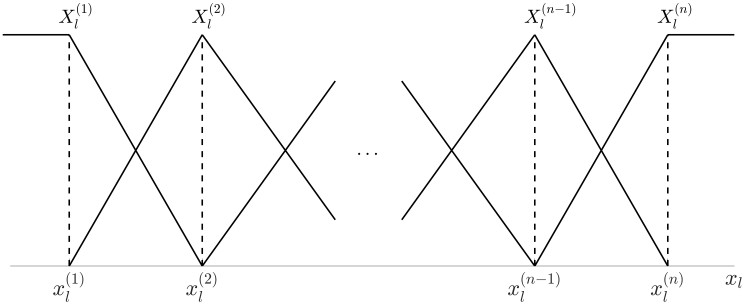

**Figure 1.** Input membership function $x_l$.

Consider the fuzzy sets $X_{n+1}^{(i_1...i_n)}$ of the output $x_{n+1}$, $\forall i_l = \{1,...,r_l\}$, $\forall l = \{1,...,n\}$, with the corresponding membership functions $\mu_{X_{n+1}^{(i_1...i_n)}}(x_{n+1})$, (see Figure 2). Where $X_{n+1}^{(i_1...i_n)}$ are their centres of gravity, and all their areas $A_{n+1}^{(i_1...i_n)}$ are the same. Then, each rule $R^{(i_1...i_n)}$ of the Mamdani model can be defined using the centres of gravity instead of the fuzzy sets for each consequent [36,37]:

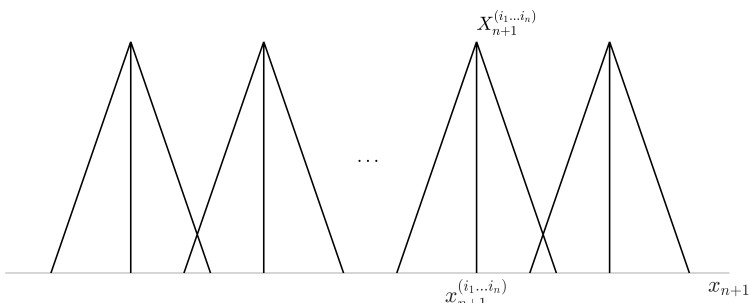

**Figure 2.** Output membership function $x_{n+1}$.

$$\text{IF } x_1 \text{ is } X_1^{(i_1)} \text{ AND ... AND } x_n \text{ is } X_n^{(i_n)} \text{ THEN } x_{n+1} = X_{n+1}^{(i_1...1_n)}, \tag{2}$$

Therefore, as more sets are used in the measurable variable, the output looks as follows:

$$x_{n+1} = \frac{\sum_{i_1=1}^{r_1} ... \sum_{i_n=1}^{r_n} w^{(i_1...i_n)}(\mathbf{x}) x_{n+1}^{(i_1...i_n)}}{\sum_{i_1=1}^{r_1} ... \sum_{i_n=1}^{r_n} w^{(i_1...i_n)}(\mathbf{x})} \tag{3}$$

where

$$w^{(i_1...i_n)}(\mathbf{x}) = \prod_{l=1}^{n} \mu_{X_l^{(i_l)}}(x_l) \tag{4}$$

### 3. Dynamic Analysis of Non-Linear Fuzzy Systems: First-Order Model

The first-order model, according to [35], is expressed as:

$$x_2 = f(x_1, u) \tag{5}$$

In the continuos case, $x_1 = x(t), x_2 = \frac{dx}{dt}$. It is assumed that $X_1^{(i_1)}$ is a fuzzy set for input $x_1$, $\forall\, i_1 = \{1, ..., r_1\}$. $r_1$ is the number of fuzzy sets for $x_1$, and $\mu_{X_1^{(i_1)}}(x_1)$ are the corresponding membership functions (see Figure 1), which are overlapped by pairs. This means,

$$\sum_{i_1=1}^{r_1} \mu_{X_1^{(i_1)}}(x_1) = 1, \quad \forall\, x_1 \tag{6}$$

For the continuous case and for a first-order model,

$$\dot{x} = f(x, u) \tag{7}$$

$$a_1 \frac{dx(t)}{dt} + a_0 x(t) = b_0 u(t), \tag{8}$$

With zero initial conditions. Rewriting the differential equation as follows:

$$\frac{a_1}{a_0} \frac{dx}{dt} + x = \frac{b_0}{a_0} u(t) \tag{9}$$

Which can be written as follows:

$$\tau \dot{x} + x = Ku \tag{10}$$

$$\frac{X_{(s)}}{U_{(s)}} = \frac{K}{\tau s + 1} \tag{11}$$

*3.1. First-Order Model*

A fuzzy dynamic model equivalent to the linear first-order dynamic model can be obtained by using the following logical operators in the fuzzy inference process; the product for conjunction, the product for implication, the sum for aggregation and the centroid for defuzzification [6,37].

Using the Mamdani model:

$$R^{(11)} : \text{IF } u \text{ is } N \text{ AND } x \text{ is } N \text{ THEN } \dot{x} = 0$$

$$R^{(12)} : \text{IF } u \text{ is } N \text{ AND } x \text{ is } P \text{ THEN } \dot{x} = -\frac{2}{\tau}$$

$$R^{(21)} : \text{IF } u \text{ is } P \text{ AND } x \text{ is } N \text{ THEN } \dot{x} = \frac{2}{\tau}$$

$$R^{(22)} : \text{IF } u \text{ is } P \text{ AND } x \text{ is } P \text{ THEN } \dot{x} = 0$$

Since the membership functions for $u$ and $x$ are:

$$\mu_N(u) = \frac{1 - Ku}{2}, \ \forall\, u \ \in \ \left[-\frac{1}{K}, \frac{1}{K}\right]$$

$$\mu_P(u) = \frac{Ku + 1}{2}, \ \forall\, u \ \in \ \left[-\frac{1}{K}, \frac{1}{K}\right]$$

$$\mu_N(x) = \frac{1 - x}{2}, \ \forall\, x \ \in \ [-1, 1]$$

$$\mu_P(x) = \frac{x + 1}{2}, \ \forall\, x \ \in \ [-1, 1]$$

The output of the system can also be computed by applying (3)

$$\sum_{i_1=1}^{2} \sum_{i_2=1}^{2} w^{(i_1 i_2)}(u,x) \dot{x}^{(i_1 i_2)} = \mu_N(u)\mu_N(x)\dot{x}^{(11)} + \mu_N(u)\mu_P(x)\dot{x}^{(12)}$$

$$+ \mu_P(u)\mu_N(x)\dot{x}^{(21)} + \mu_P(u)\mu_P(x)\dot{x}^{(22)}$$

$$= \left(\frac{1-Ku}{2}\right)\left(\frac{1-x}{2}\right)(0) + \left(\frac{1-Ku}{2}\right)\left(\frac{x+1}{2}\right)\left(-\frac{2}{\tau}\right) \quad (12)$$

$$+ \left(\frac{Ku+1}{2}\right)\left(\frac{1-x}{2}\right)\left(\frac{2}{\tau}\right) + \left(\frac{Ku+1}{2}\right)\left(\frac{x+1}{2}\right)(0)$$

$$= -\frac{x}{\tau} + \frac{Ku}{\tau}$$

where:

$$\sum_{i_1=1}^{2} \sum_{i_2=1}^{2} w^{(i_1 i_2)}(u,x) = \mu_N(u)\mu_N(x) + \mu_N(u)\mu_P(x)$$

$$+ \mu_P(u)\mu_N(x) + \mu_P(u)\mu_P(x)$$

$$= \left(\frac{1-Ku}{2}\right)\left(\frac{1-x}{2}\right) + \left(\frac{1-Ku}{2}\right)\left(\frac{x+1}{2}\right) \quad (13)$$

$$+ \left(\frac{Ku+1}{2}\right)\left(\frac{1-x}{2}\right) + \left(\frac{Ku+1}{2}\right)\left(\frac{x+1}{2}\right)$$

$$= 1$$

Which matches Equation (10).

$$\dot{x} = -\frac{x}{\tau} + \frac{Ku}{\tau} \quad (14)$$

Input and output scaling factors [38] will facilitate model fitting by modifying $K, \tau$. Thus, the universe of discourse will be normalised within the interval $[-1, 1]$, as shown in Figure 3.

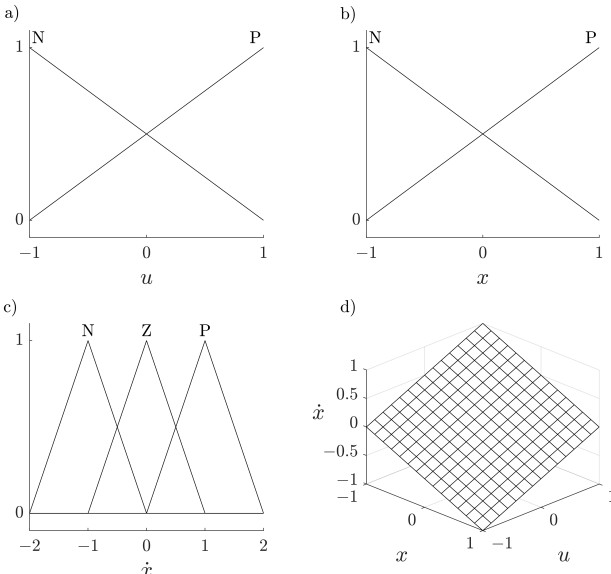

**Figure 3.** Fuzzy sets; (**a**) input set $u$, (**b**) input set $x$, (**c**) output set $\dot{x}$ centre of gravity and (**d**) output surface between $\dot{x}$, $x$ and $u$.

The number of sets used in the input can be modified by applying scaling factors. For example, from 2 to 3 fuzzy sets, as shown in the output inference map in Figure 4 below.

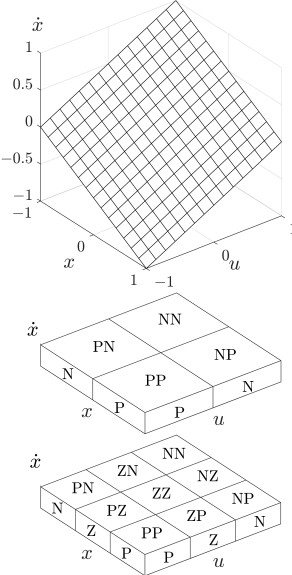

**Figure 4.** First-order inference map.

The scaling factors will be as follows for the input $u = K$, at the output $\dot{x} = \frac{2}{\tau}$ (see Figure 5).

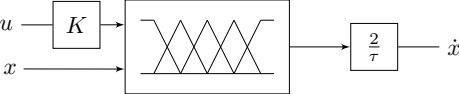

**Figure 5.** Scaling factors for the first-order system.

After obtaining an equivalent first-order fuzzy model, it is possible to modify the triangular function with linguistic hedges and thus adjust the obtained model. Usually, the model is adjusted by replacing the triangular membership functions with existing membership functions.

### 3.2. Linguistic Hedges in Model

A linguistic variable carries with it the concept of fuzzy set qualifiers, called hedges. The linguistic variables describe phenomena that are too complex or poorly defined to be described in quantitative terms.

Traditionally, modifiers of fuzzy sets have been used, which are called linguistic labels, which in natural language, are equivalent to adverbs. Interpreting these statements in the fuzzy model requires the composition of the membership function with a simple arithmetic operation [15].

As a special case of $\mu_A(x) = \mu_A^p(x)$ when $p > 1$, the operation of concentration (very) can be described as follows:

$$\mu_{CON(A)}(x) = \mu_A^2(x) \ \forall\, x \in X$$

It is common to consider the interpretation of the adverb "very" as the square of the original membership function, i.e., "The system is very fast" would be interpreted as:

$$\mu_{Very\ Fast}(x) = \mu_{VF}(x) = (\mu_{Fast}(x))^2$$

The opposite case of $\mu_A(x) = \mu_A^p(x)$ when $0 < p < 1$, is the dilation (more-or-less) expressed by

$$\mu_{DIL(A)}(x) = \mu_A^{0.5}(x) \ \forall \, x \in \, X$$

The interpretation of the adverb "more-or-less" would be implemented as the square root of the membership function. It would be interpreted as:

$$\mu_{More-or-less \ Fast}(x) = \mu_{MF}(x) = (\mu_{Fast}(x))^{0.5}$$

### 3.3. First-Order Fuzzy Model with Linguistic Hedges

A model with triangular membership functions is developed, which will now be modified with the linguistic hedges type mentioned above, such as dilation and concentration. We will use a non-linear model that is represented in this way: $\dot{x} = -\frac{x|x|}{\tau} + \frac{K}{\tau}u$, in a fuzzy system [39]. It will be a concentration operation $\mu_X{}^2(x)$, and it is shown in Figure 6. If the model is $\dot{x} = -\frac{\sqrt{|x|}}{\tau} + \frac{K}{\tau}u$ [37], this represents a dilation $\mu_X{}^{0.5}(x)$ (see Figure 7). The linguistic hedges used for the concentration at input x will be $X_1^{(1)} = VN$, $X_1^{(2)} = VZ$, $X_1^{(3)} = VP$. The dilation of x is expressed by linguistic hedges $X_1^{(1)} = MN$, $X_1^{(2)} = MZ$, $X_1^{(3)} = MP$.

The resultant Mamdani model:

$$R^{(11)} : \text{IF } u \text{ is } N \text{ AND } x \text{ is } X_1^{(1)} \text{ THEN } \dot{x} = 0$$

$$R^{(12)} : \text{IF } u \text{ is } Z \text{ AND } x \text{ is } X_1^{(1)} \text{ THEN } \dot{x} = 0.5$$

$$R^{(13)} : \text{IF } u \text{ is } P \text{ AND } x \text{ is } X_1^{(1)} \text{ THEN } \dot{x} = 1$$

$$R^{(21)} : \text{IF } u \text{ is } N \text{ AND } x \text{ is } X_1^{(2)} \text{ THEN } \dot{x} = -0.5$$

$$R^{(22)} : \text{IF } u \text{ is } Z \text{ AND } x \text{ is } X_1^{(2)} \text{ THEN } \dot{x} = 0$$

$$R^{(23)} : \text{IF } u \text{ is } P \text{ AND } x \text{ is } X_1^{(2)} \text{ THEN } \dot{x} = 0.5$$

$$R^{(31)} : \text{IF } u \text{ is } N \text{ AND } x \text{ is } X_1^{(3)} \text{ THEN } \dot{x} = -1$$

$$R^{(32)} : \text{IF } u \text{ is } Z \text{ AND } x \text{ is } X_1^{(3)} \text{ THEN } \dot{x} = -0.5$$

$$R^{(33)} : \text{IF } u \text{ is } P \text{ AND } x \text{ is } X_1^{(3)} \text{ THEN } \dot{x} = 0$$

The scaling factors for the concentration and dilation will be as follows for the input $u = K$ valid for $0 \le K \le 1$, at the output $\dot{x} = \frac{2}{\tau}$ (see Figure 5).

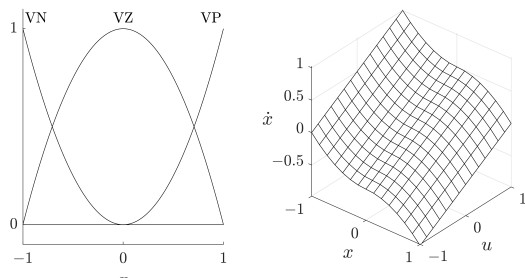

**Figure 6.** The concentration of the input membership functions x and the resulting inference map.

The response of a unit step using linguistic operators, concentration and dilation are compared to a first-order linear system in Figure 8. This graph indicates that the settling time ($t_s$) is less than that of the first-order linear system in concentration, which implies that it is faster than the first-order linear system. On the contrary, the $t_s$ in dilation is higher than that of the first-order linear system. Therefore, the system is slower compared to the

first-order system. This is valid only for the first 10 s. After that, it is observed that the dilation is faster than the concentration.

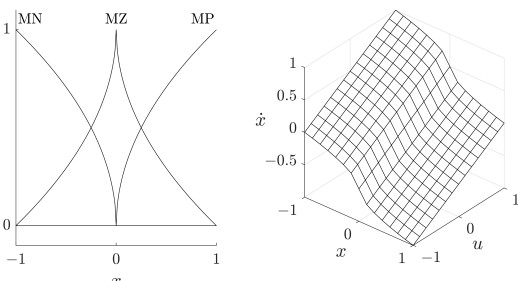

**Figure 7.** The dilation of the input membership functions x and the resultant inference map.

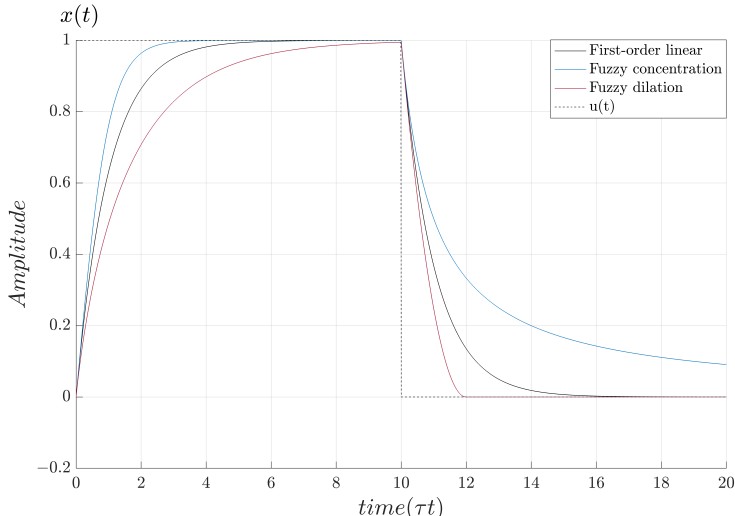

**Figure 8.** Step response, with normalised time $\tau t$, for K = 1.

### 3.4. Dynamic Characteristics of a First-Order Fuzzy System

Once the model has been adjusted, a dynamic analysis is performed on this first-order system for input $u$, and there are two fuzzy sets for $u_1$, $U^{i_0}$, $\forall\, i_0 = \{1, 2\}$, as shown in Figure 9.

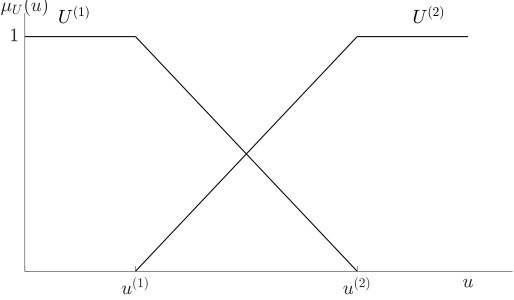

**Figure 9.** Triangular membership functions for the reference input $u$.

The input of a step starts from a reference point $u^{(1)}$ to another $u^{(2)}$, remaining at a final value of $u^{(2)}$. To reach the steady-state regime, it is theoretically required that $t \to \infty$. However, it is important to determine a practical value of the time at which the system reaches its final value.

It is assumed that the system is initially in equilibrium, i.e., $x_1 = x_1^{(1)}$ and $x_2 = x_2^{(1)}$. Then, according to [35], the settling time of $x_2(t)$ when $x_1$ becomes $x_1 = x_1^{(2)}$ will have the following dynamics:

$$x_2 = \frac{dx_1}{dt}, \quad then \quad dt = \frac{dx_1}{x_2} \tag{15}$$

To calculate the settling time, we will use the criterion of 95% of the final value so that the response reaches its final value as performed in (15). There are $r_1$ fuzzy sets, as shown in Figure 1. This can be calculated as the response to a step input where the system response is part of a reference point $x_1^i = Ku^i$ to another $x_1^f = Ku^f$, remaining in a final value $x_1^f = Ku^f$. $t = 0$ is defined to be the beginning of the response to the step input and ends at $t = t_s$ to obtain the settling time. The limits of the integral will be,

$$t_s = \int_0^{t_s} dt = \int_{x_1^i}^{x_1^i + 0.95(x_1^f - x_1^i)} \frac{1}{x_2} \, dx_1 \tag{16}$$

When a system has a delay, the dynamics of the system will have the same behaviour in the transient response. It will also be settled at the same steady-state value. The only change is that the system will have a time delay $(t_d)$ responding once the input signal is applied to the system. In this case, a step input is defined as $t = t_d$ when the response to the step input begins and ends at $t = t_s$ to calculate the settling time. The limits of the integral will be,

$$t_s = \int_{t_d}^{t_s} dt = \int_{x_1^i}^{x_1^i + 0.95(x_1^f - x_1^i)} \frac{1}{x_2} \, dx_1 \tag{17}$$

## 4. Dynamic Analysis of Non-Linear Fuzzy Systems: Second-Order Model

This section aims to provide a fuzzy dynamic system analysis. This work will be extended to the technique of fuzzy non-linear analysis. Theoretical analysis is often the most efficient way to explore the characteristics of a system. A continuous system can be represented by differential equations using the values of the input and output variables. For the second-order case of three input variables and one output variable, such an equation will be obtained as follows:

$$f(x_1, x_2, u) = x_3 \tag{18}$$

System variables are defined as follows:

$$x_1 = x(t), \quad x_2 = \frac{dx}{dt}, \quad x_3 = \frac{d^2 x}{dt^2} \tag{19}$$

Often, the input $u$ is zero because the system is stationary or time-invariant.

Reviewing the literature, one of the graphical methods used is phase-plane analysis. This plane is two-dimensional and is helpful for second-order systems.

Although the simulation is important for non-linear control, it must be supported by theoretical analysis. Blind simulation of non-linear systems is likely to yield few or erroneous results. This is especially true because of the different behaviours that non-linear systems can exhibit.

$$\ddot{x} = f(x, \dot{x}, u) \tag{20}$$

$$a_2 \frac{d^2 x(t)}{dt^2} + a_1 \frac{dx(t)}{dt} + a_0 x(t) = b_0 u(t), \tag{21}$$

With zero initial conditions. As with first-order systems, a standard form has been established. Rewriting the differential equation as follows:

$$\frac{a_2}{a_0} \frac{d^2 x}{dt^2} + \frac{a_1}{a_0} \frac{dx}{dt} + x = \frac{b_0}{a_0} u(t) \tag{22}$$

Two parameters can be used to describe the characteristics of the second-order transient response in the same way that the time constant of the first-order system is determined. The two quantities are known as the natural frequency and the relative damping factor. The resultant transfer function becomes:

$$\frac{X_{(s)}}{U_{(s)}} = \frac{K}{\frac{s^2}{\omega_n^2} + \frac{2\zeta s}{\omega_n} + 1} = \frac{K\omega_n^2}{s^2 + 2\zeta\omega_n s + \omega_n^2} \tag{23}$$

*4.1. Second-Order Approach*

A fuzzy dynamic model equivalent to the linear second-order dynamic model can be obtained by using the following logical operators in the fuzzy inference process; the product for conjunction, the product for implication, the sum for aggregation and the centroid for defuzzification. These operators are used so that the output of the fuzzy system is linear. In the second-order model, we obtain that $\ddot{x}^{(1)} = 8K\omega_n^2$, $\ddot{x}^{(2)} = 0$, $\ddot{x}^{(3)} = 0$, $\ddot{x}^{(4)} = -8K\omega_n^2$, $\ddot{x}^{(5)} = 4K\omega_n^2$ and $\ddot{x}^{(6)} = -4K\omega_n^2$ can be approximated. Another important thing to consider is the universe of discourse, which can be obtained from the phase plane.
Using the Mamdani model:

$$R^{(1)} : \text{IF } x \text{ is } N \text{ AND } \dot{x} \text{ is } N \text{ THEN } \ddot{x} = 8K\omega_n^2$$

$$R^{(2)} : \text{IF } x \text{ is } N \text{ AND } \dot{x} \text{ is } P \text{ THEN } \ddot{x} = 0$$

$$R^{(3)} : \text{IF } x \text{ is } P \text{ AND } \dot{x} \text{ is } N \text{ THEN } \ddot{x} = 0$$

$$R^{(4)} : \text{IF } x \text{ is } P \text{ AND } \dot{x} \text{ is } P \text{ THEN } \ddot{x} = -8K\omega_n^2$$

$$R^{(5)} : \text{IF } u \text{ is } P \text{ THEN } \ddot{x} = 4K\omega_n^2$$

$$R^{(6)} : \text{IF } u \text{ is } N \text{ THEN } \ddot{x} = -4K\omega_n^2$$

These rules proved that the fuzzy system is equivalent to a linear system.
Since the membership functions for $x$ and $\dot{x}$ are:

$$\mu_N(x) = \frac{2K - x}{4K}, \ \forall \, x \, \in \, [-2K, 2K]$$

$$\mu_P(x) = \frac{x + 2K}{4K}, \ \forall \, x \, \in \, [-2K, 2K]$$

$$\mu_N(\dot{x}) = \frac{K\omega_n - \zeta\dot{x}}{2K\omega_n}, \ \forall \, \dot{x} \, \in \, \left[ -\frac{K\omega_n}{\zeta}, \frac{K\omega_n}{\zeta} \right]$$

$$\mu_P(\dot{x}) = \frac{\zeta\dot{x} + K\omega_n}{2K\omega_n}, \ \forall \, \dot{x} \, \in \, \left[ -\frac{K\omega_n}{\zeta}, \frac{K\omega_n}{\zeta} \right]$$

$$\mu_N(u) = \frac{2 - u}{4}, \ \forall \, u \, \in \, [-2, 2]$$

$$\mu_P(u) = \frac{u + 2}{4}, \ \forall \, u \, \in \, [-2, 2]$$

The output of the system can be computed as follows:

$$\ddot{x} = \frac{\sum_{i_1=1}^{6} w^{(i_1)}(x) \cdot \ddot{x}^{(i_1)}}{\sum_{i_1=1}^{6} w^{(i_1)}(x)} \tag{24}$$

by applying

$$
\begin{aligned}
\sum_{i_1=1}^{6} \mathrm{w}^{(i_1)}(x)\ddot{x}^{(i_1)} &= \mu_N(x)\mu_N(\dot{x})\ddot{x}^{(1)} + \mu_N(x)\mu_P(\dot{x})\ddot{x}^{(2)} \\
&\quad + \mu_P(x)\mu_N(\dot{x})\ddot{x}^{(3)} + \mu_P(x)\mu_P(\dot{x})\ddot{x}^{(4)} \\
&\quad + \mu_P(u)\ddot{x}^{(5)} + \mu_N(u)\ddot{x}^{(6)} \\
&= \left(\frac{2K-x}{4K}\right)\left(\frac{K\omega_n - \zeta\dot{x}}{2K\omega_n}\right)(8K\omega_n^2) + \left(\frac{2K-x}{4K}\right)\left(\frac{\zeta\dot{x}+K\omega_n}{2K\omega_n}\right)(0) \\
&\quad + \left(\frac{x+2K}{4K}\right)\left(\frac{K\omega_n-\zeta\dot{x}}{2K\omega_n}\right)(0) + \left(\frac{x+2K}{4K}\right)\left(\frac{\zeta\dot{x}+K\omega_n}{2K\omega_n}\right)(-8K\omega_n^2) \\
&\quad + \left(\frac{u+2}{4}\right)(4K\omega_n^2) + \left(\frac{2-u}{4}\right)(-4K\omega_n^2) \\
&= -4\zeta\omega_n\dot{x} - 2\omega_n^2 x + 2K\omega_n^2 u
\end{aligned}
\tag{25}
$$

where:

$$
\begin{aligned}
\sum_{i_1=1}^{6} \mathrm{w}^{(i_1)}(x) &= \mu_N(x)\mu_N(\dot{x}) + \mu_N(x)\mu_P(\dot{x}) \\
&\quad + \mu_P(x)\mu_N(\dot{x}) + \mu_P(x)\mu_P(\dot{x}) \\
&\quad + \mu_P(u) + \mu_N(u) \\
&= \left(\frac{2K-x}{4K}\right)\left(\frac{K\omega_n - \zeta\dot{x}}{2K\omega_n}\right) + \left(\frac{2K-x}{4K}\right)\left(\frac{\zeta\dot{x}+K\omega_n}{2K\omega_n}\right) \\
&\quad + \left(\frac{x+2K}{4K}\right)\left(\frac{K\omega_n-\zeta\dot{x}}{2K\omega_n}\right) + \left(\frac{x+2K}{4K}\right)\left(\frac{\zeta\dot{x}+K\omega_n}{2K\omega_n}\right) \\
&\quad + \left(\frac{u+2}{4}\right) + \left(\frac{2-u}{4}\right) \\
&= 2
\end{aligned}
\tag{26}
$$

Which matches the polynomial in Equation ((23))

$$
\ddot{x} = -2\zeta\omega_n\dot{x} - \omega_n^2 x + K\omega_n^2 u
\tag{27}
$$

To allow the adjustments of $K, \omega_n$ and $\zeta$, input and output scaling factors will be used. For this purpose, the universe of discourse will be left normalised to the interval $[-1, 1]$ (see Figure 10). For the output, the centre of gravity will be $\{-1, -0.5, 0, 0.5, 1\}$ and will be multiplied by the maximum value of the output. In this case, it becomes $8K\omega_n^2$. The input variable $x$, which is within the interval $[-2K, 2K]$, will be normalised to $\frac{1}{2K}$, and then the variable $\dot{x}$ will be normalised to $\frac{\zeta}{K\omega_n}$. Next, $u$ will be normalised to $\frac{1}{2}$.

The number of sets used in the input can be modified by applying scaling factors. For example, from 2 to 3 fuzzy sets, as shown in the output inference map in Figure 11 below.

The scaling factors will be as follows for the inputs $u = \frac{1}{2}$, $x = \frac{1}{2K}$ and $\dot{x} = \frac{\zeta}{K\omega_n}$ at the output $\ddot{x} = 8K\omega_n^2$ (see Figure 12).

It is possible to modify the triangular function with linguistic hedges and thus adjust the obtained model. Most models are adjusted by replacing triangular membership functions with existing membership functions.

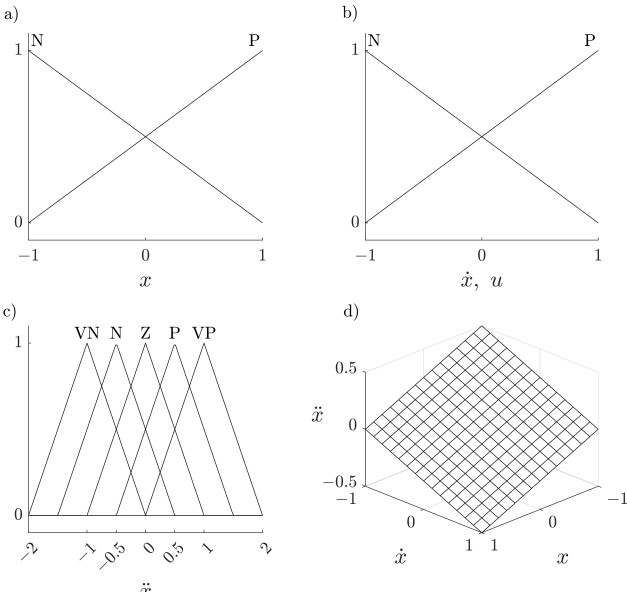

**Figure 10.** Fuzzy sets; (**a**) input set $x$, (**b**) input set $\dot{x}$, $u$, (**c**) output set $\ddot{x}$ centre of gravity and (**d**) output surface between $\ddot{x}$, $\dot{x}$ and $x$.

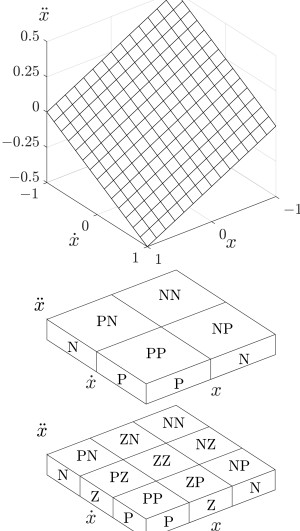

**Figure 11.** Second-order inference map.

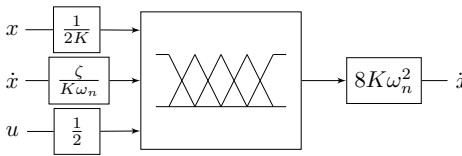

**Figure 12.** Scaling factors for a second-order system.

### 4.2. Second-Order Fuzzy Model with Linguistic Hedges

A model with triangular membership functions was developed, which will now be modified with the linguistic hedges type mentioned above, such as dilation and concentration. We will use a non-linear model that is represented in this way: $\ddot{x} = -2\zeta\omega_n\dot{x} - \omega_n^2 x|x| + K\omega_n^2 u$, in a fuzzy system. It will be a concentration operation $\mu_X{}^2(x)$, and it is shown in Figure 13. If the model is $\ddot{x} = -2\zeta\omega_n\dot{x} - \omega_n^2\sqrt{|x|} + K\omega_n^2 u$, this

represents a dilation $\mu_X^{0.5}(x)$ (see Figure 14). The linguistic hedges used for the concentration at input x will be $X_1^{(1)} = VN$, $X_1^{(2)} = VZ$, $X_1^{(3)} = VP$. The dilation of x is expressed by linguistic hedges $X_1^{(1)} = MN$, $X_1^{(2)} = MZ$, $X_1^{(3)} = MP$.

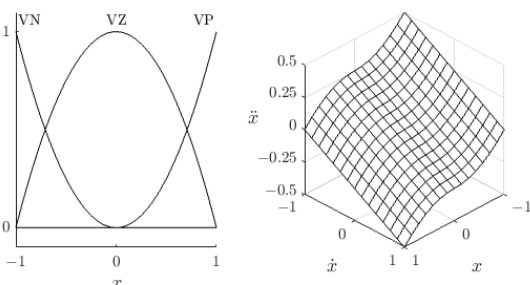

**Figure 13.** The concentration of the input membership functions x and the resulting inference map.

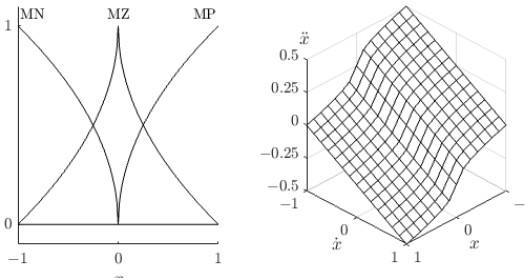

**Figure 14.** The dilation of the input membership functions x and the resultant inference map.

The resultant Mamdani model:

$$R^{(1)} \ : \text{IF } x \text{ is } X_1^{(1)} \text{ AND } \dot{x} \text{ is } N \text{ THEN } \ddot{x} = 1$$

$$R^{(2)} \ : \text{IF } x \text{ is } X_1^{(1)} \text{ AND } \dot{x} \text{ is } Z \text{ THEN } \ddot{x} = 0.5$$

$$R^{(3)} \ : \text{IF } x \text{ is } X_1^{(1)} \text{ AND } \dot{x} \text{ is } P \text{ THEN } \ddot{x} = 0$$

$$R^{(4)} \ : \text{IF } x \text{ is } X_1^{(2)} \text{ AND } \dot{x} \text{ is } N \text{ THEN } \ddot{x} = 0.5$$

$$R^{(5)} \ : \text{IF } x \text{ is } X_1^{(2)} \text{ AND } \dot{x} \text{ is } Z \text{ THEN } \ddot{x} = 0$$

$$R^{(6)} \ : \text{IF } x \text{ is } X_1^{(2)} \text{ AND } \dot{x} \text{ is } P \text{ THEN } \ddot{x} = -0.5$$

$$R^{(7)} \ : \text{IF } x \text{ is } X_1^{(3)} \text{ AND } \dot{x} \text{ is } N \text{ THEN } \ddot{x} = 0$$

$$R^{(8)} \ : \text{IF } x \text{ is } X_1^{(3)} \text{ AND } \dot{x} \text{ is } Z \text{ THEN } \ddot{x} = -0.5$$

$$R^{(9)} \ : \text{IF } x \text{ is } X_1^{(3)} \text{ AND } \dot{x} \text{ is } P \text{ THEN } \ddot{x} = -1$$

$$R^{(10)} : \text{IF } u \text{ is } N \text{ THEN } \ddot{x} = -0.5$$

$$R^{(11)} : \text{IF } u \text{ is } Z \text{ THEN } \ddot{x} = 0$$

$$R^{(12)} : \text{IF } u \text{ is } P \text{ THEN } \ddot{x} = 0.5$$

The scaling factors for the concentration will be as follows for the inputs $x = \frac{1}{(4K)^{0.5}}$, $\dot{x} = \frac{\zeta}{2K\omega_n}$ and $u = \frac{1}{4}$, at the output $\ddot{x} = 16K\omega_n^2$.

The scaling factors for the dilation will be as follows for the inputs $x = \frac{1}{(2K)^2}$, $\dot{x} = \frac{\zeta}{K\omega_n}$ and $u = \frac{1}{2}$, at the output $\ddot{x} = 8K\omega_n^2$.

The response of a unit step using linguistic operators, concentration and dilation is compared to a second-order linear system in Figure 15. This graph indicates that the peak

time (tp) is less than that of the second-order linear system in concentration, which implies that it is faster than the second-order linear system. On the contrary, the tp value is higher in dilation than that of the second-order linear system. Therefore, the system is slower compared to the second-order linear system. This is valid only for the first 20 s. After that, it is observed that the dilation is faster than the concentration.

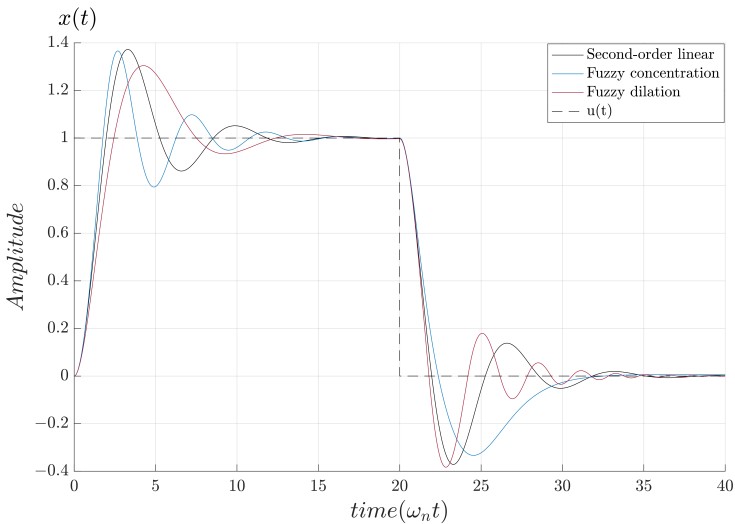

**Figure 15.** Step response, with normalised time $\omega_n t$, for K = 1 and $\zeta = 0.3$.

### 4.3. Performance of Fuzzy Second-Order Dynamical Systems

Standard performance measures are usually defined in terms of the step response of a system, as shown in Figure 15. The speed response depends on the peak time (tp). The similarity between the transient response and the step input is measured by the per cent of overshoot (Mp). The transient response of the system can be described by two factors: the speed of the response and the peak time. The closeness of the response to the desired one is represented by the overshoot and settling time [40].

In this case, explicit dynamic specification, as Mp and tp, cannot be obtained as in the first-order case. A significant number of simulations to obtain the relationship between Mp, tp and $\zeta$ are performed. This will be applied to concentration and dilation, as seen previously. In Figures 16 and 17, the per cent overshoot is shown as a function of the relative damping ratio, $\zeta$. In addition, Figures 16 and 17 present the per cent overshoot and peak time as a function of the relative damping ratio. The dotted line indicates the performance of the second-order linear system, while the solid one corresponds to the non-linear system. As shown in Figure 16, the segmented line shows the concentration of a fuzzy system, while in Figure 17, the dilation of a fuzzy system is shown. This analysis is valid for dynamical systems; undamped, underdamped and critically damped. In the case of overdamped, it has no peak time and per cent overshoot.

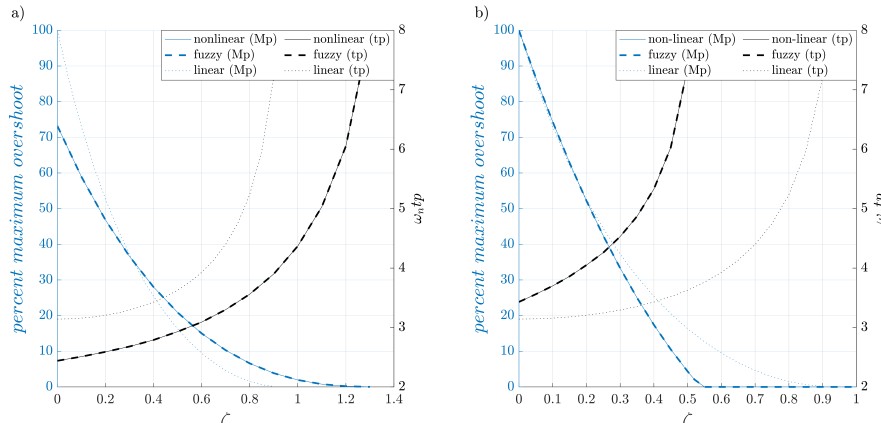

**Figure 16.** The result for concentration graphs shows the per cent overshoot and normalised peak time as a function of damping ratio $\zeta$; (**a**) step-up, (**b**) step-down.

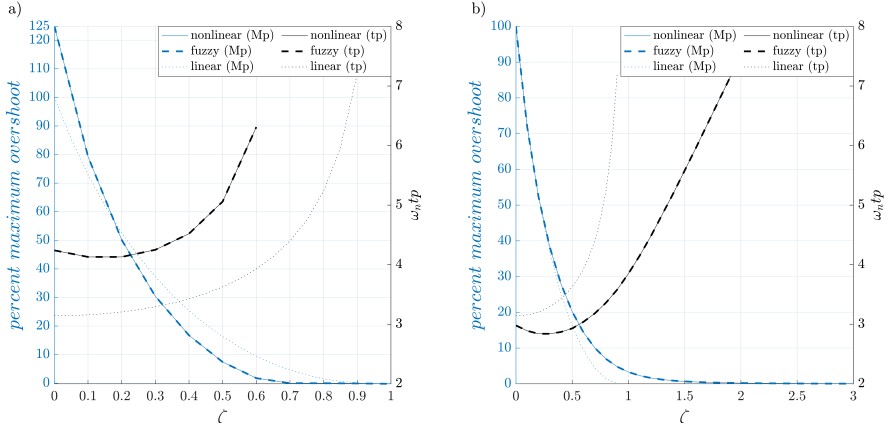

**Figure 17.** The result for dilation graphs shows the per cent overshoot and normalised peak time as a function of damping ratio $\zeta$; (**a**) step-up, (**b**) step-down.

## 5. Application

To explain what we have studied above, we will apply the concepts to the first-order system of a tank and calculate the settling time. The analysis of the second-order system will be applied to a simple pendulum.

### 5.1. Model of the Tank

The GÜNT RT450.01 consists of a transparent plastic level tank as the main element and uses water as the working medium. It provides the following features: section tank $A = 0.0147$ m² with a liquid of height $h(t)$, volume $v(t)$ and outlet flow $q_o(t)$ due to an input flow $q_i(t)$. It presents a delay time of $t_d = 10$ s and a percentage of valve opening $\alpha(t) = 0.5$, with a maximum output section $C = 2.83 \times 10^{-5}$ m² and gravitational acceleration $g = 9.81$ m/s², as shown in Figure 18.

An empirical model of this multivariate system is given by

$$\dot{v}(t) = q_i(t - t_d) - q_o(t)$$
$$v(t) = Ah(t) \tag{28}$$
$$q_o(t) = C\alpha(t)\sqrt{2gh(t)}$$

$$\dot{v}(t) = q_i(t - t_d) - q_o(t) \tag{29}$$

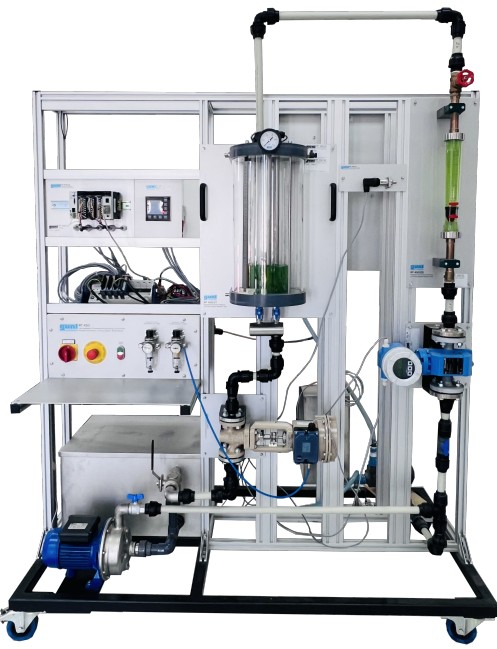

**Figure 18.** GÜNT RT450.01 level tank module.

This equation represents a multivariate relationship between $q_i(t)$, $\alpha(t)$ and $h(t)$:

$$A\dot{h}(t) = q_i(t - t_d) - C\alpha(t)\sqrt{2gh(t)} \tag{30}$$

Rewriting in the standard form,

$$\frac{A\,\dot{h}(t)}{C\alpha(t)\sqrt{2g}} + \sqrt{h(t)} = \frac{q_i(t - t_d)}{C\alpha(t)\sqrt{2g}} \tag{31}$$

The static gain of this non-linear system is when the derivative is zero, it is as follows:

$$\sqrt{h(t)} = \frac{q_i(t - t_d)}{C\alpha(t)\sqrt{2g}} \tag{32}$$

Therefore, for $t > T_d$ the static gain K would be,

$$h = K = \left(\frac{q_i}{C\alpha(t)\sqrt{2g}}\right)^2 \tag{33}$$

and the model for this condition is,

$$\frac{A\,\dot{h}(t)}{C\alpha\sqrt{2g}} + \sqrt{h(t)} = \frac{q_i(t)}{C\alpha\sqrt{2g}} \tag{34}$$

The initial conditions to fill the tank are $q_i^{(0)} = 1.09 \times 10^{-5}$ m³/s. Replaced in (33) gives $h^{(0)} = 0.03$ m. To analyse the dynamic behaviour of this empirical model, initially, the tank is in equilibrium at $h^{(0)} = 0.03$ m; at a time of 35 s, the inflow increases to $q_i = 3.1 \times 10^{-5}$ m³/s and then after 628 s, it drops to $q_i = 0$ m³/s.

It can be seen that the system is in a steady state and has a final value of around $K = 0.244$ m. Figure 19 shows the dynamic evolution of the system, with a sampling time of 0.5 s.

The Mamdani model of the tank model in the standard form (Equation (34)) can be approximated; the normalised fuzzy sets between [0, 1] are shown in Figure 20.

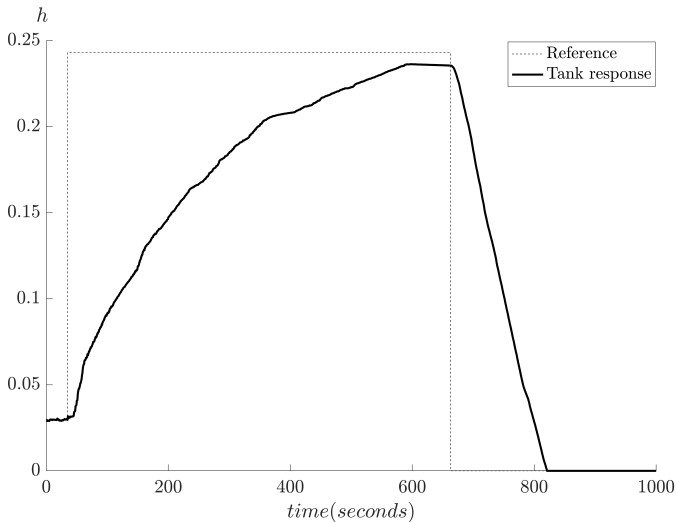

**Figure 19.** Dynamical evolution of the tank.

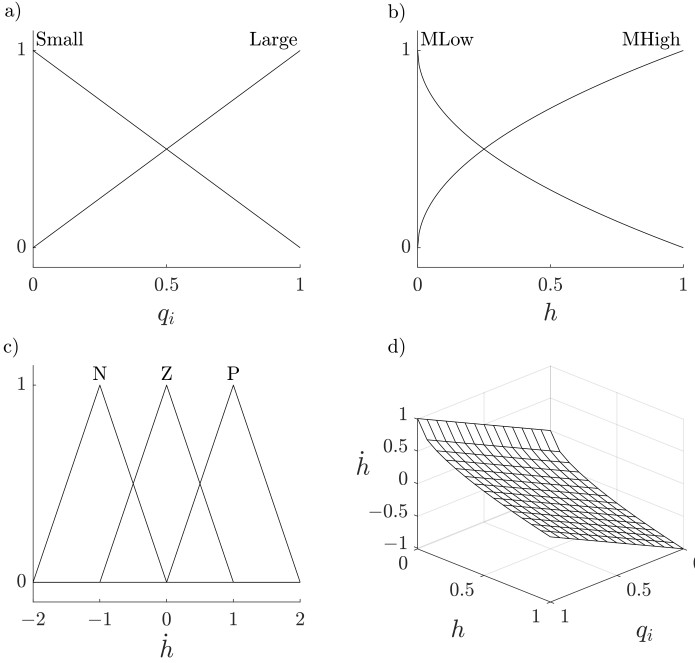

**Figure 20.** Fuzzy sets tank; (**a**) input set $q_i$, (**b**) input set $h$, (**c**) output set $\dot{h}$ centre of gravity and (**d**) output surface between $\dot{h}$, $h$ and $q_i$.

Using the input and output scaling factors, the delay time was added in Figure 21.

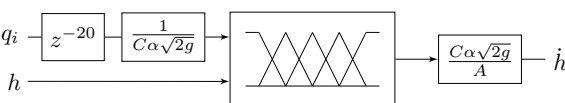

**Figure 21.** Input and output scaling factors.

The resultant Mamdani model becomes:

$$R^{(11)} : \text{IF } q_i \text{ is } Small \text{ AND } h \text{ is } MLow \text{ THEN } \dot{h} = 0$$

$$R^{(12)} : \text{IF } q_i \text{ is } Small \text{ AND } h \text{ is } MHigh \text{ THEN } \dot{h} = -\frac{C\alpha\sqrt{2g}}{A}$$

$$R^{(21)} : \text{IF } q_i \text{ is } Large \text{ AND } h \text{ is } MLow \text{ THEN } \dot{h} = \frac{C\alpha\sqrt{2g}}{A}$$

$$R^{(22)} : \text{IF } q_i \text{ is } Large \text{ AND } h \text{ is } MHigh \text{ THEN } \dot{h} = 0$$

$$(35)$$

The membership functions for $q_i$ and $h$ applying the dilation are:

$$\mu_{Small}(q_i) = 1 - \frac{q_i}{C\alpha\sqrt{2g}}, \quad \forall\, q_i \in [0, C\alpha\sqrt{2g}]$$

$$\mu_{Large}(q_i) = \frac{q_i}{C\alpha\sqrt{2g}}, \quad \forall\, q_i \in [0, C\alpha\sqrt{2g}]$$

$$\mu_{MLow}(h) = 1 - \sqrt{h}, \quad \forall\, h \in [0,1]$$

$$\mu_{MHigh}(h) = \sqrt{h}, \quad \forall\, h \in [0,1]$$

Then the output is as follows:

$$\dot{h} = \sum_{i_1=1}^{2}\sum_{i_2=1}^{2} \text{w}^{(i_1 i_2)}(h, q_i)\dot{h}^{(i_1 i_2)} =$$

$$\mu_{Small}(q_i)\mu_{MLow}(h)\dot{h}^{(11)} + \mu_{Small}(q_i)\mu_{MHigh}(h)\dot{h}^{(12)}$$

$$\mu_{Large}(q_i)\mu_{MLow}(h)\dot{h}^{(21)} + \mu_{Large}(q_i)\mu_{MHigh}(h)\dot{h}^{(22)}$$

$$\dot{h} = \left(1 - \frac{q_i}{C\alpha\sqrt{2g}}\right)(1 - \sqrt{h})(0) + \left(1 - \frac{q_i}{C\alpha\sqrt{2g}}\right)(\sqrt{h})\left(-\frac{C\alpha\sqrt{2g}}{A}\right)$$

$$+ \left(\frac{q_i}{(C\alpha\sqrt{2g})}\right)(1 - \sqrt{h})\left(\frac{C\alpha\sqrt{2g}}{A}\right) + \left(\frac{q_i}{C\alpha\sqrt{2g}}\right)(\sqrt{h})(0)$$

$$(36)$$

$$\dot{h}(t) = \frac{q_i(t)}{A} - \frac{C\alpha\sqrt{2g}}{A}\sqrt{h(t)} \tag{37}$$

This model coincides with the tank model in the standard form (Equation (34)). The fuzzy model output is shown in Figure 22.

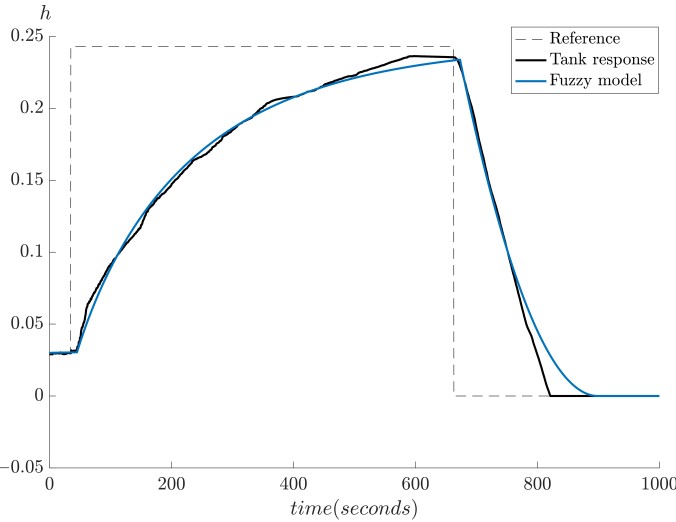

**Figure 22.** Dynamical evolution of the tank and fuzzy model.

To calculate the settling time for the step from $q_i^i = 1.09 \times 10^{-5}$ to $q_i^f = 3.1 \times 10^{-5}$ m$^3$/s, we replace both of them in (33) to obtain a height of $h^i = 0.03$ m, up to $h^f = 0.244$ m, respectively. Then (Equation (17)) is used to calculate the settling time.

$$\int_{10}^{t_s} dt = \int_{0.03}^{0.233} \frac{1}{2.11 \cdot 10^{-3} - 4.26 \cdot 10^{-3}\sqrt{h}}\, dh$$

The calculated settling time is $t_s = 618$ s.

To calculate the settling time for the step-down where the level is initially set as $h^i = 0.235$ m, the inlet flow, $q_i = 0$, is replaced in (33) to give us a height of $h^f = 0$ m. Equation (17) is then used to calculate the settling time.

$$\int_{10}^{t_s} dt = \int_{0.235}^{0.012} \frac{1}{-4.26 \cdot 10^{-3}\sqrt{h}}\, dh$$

The calculated settling time is $t_s = 186$ s.

The performance of different models is compared. Time delays in the identification of processes can have a significant impact. For systems with time delays, there are several models proposed. The non-linear ARX (NLARX) model is a flexible non-linearity estimator with parameters that do not need to have physical significance, commonly used in time series modelling. This structure employs a wavelet or sigmoid function to model complex non-linear behaviour [41]. NLARX models can also be estimated using MATLAB System Identification Toolbox. As shown in Figure 23, with the fuzzy model (in blue) has the best performance, with 92.91%, followed by the NLARX models (in green), with a performance of 90.16% and, finally, the first-order model (in grey) with a performance of 77.77%.

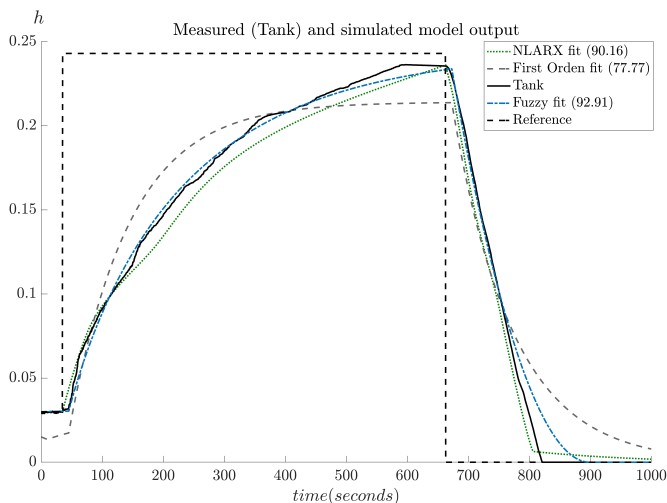

**Figure 23.** Performance comparison of different models.

### 5.2. Simple Pendulum Model

In this example, the second-order fuzzy model described in Section 4.1 is used to model a simple pendulum. A normalised sinusoidal function modifies the membership function of the angle $\theta$ to demonstrate that, by an appropriate choice of membership function, the model is identical to the one obtained mathematically.

A practical example of the dynamic model of a simple pendulum is provided. The pendulum's differential equation is:

$$J\ddot{\theta}(t) + B\dot{\theta}(t) + mgl\sin(\theta(t)) = \tau(t) \tag{38}$$

When $J = ml^2$ is the inertia of the pendulum+ball set, B is the friction on the axis of rotation, $l$ is the length of the pendulum and $\tau(t)$ is the applied torque. See Figure 24.

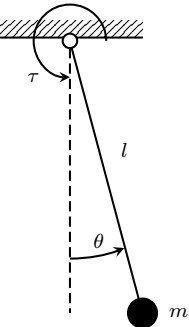

**Figure 24.** Simple pendulum.

If it is written in standard form as (22), the pendulum model remains;

$$\frac{l}{g}\ddot{\theta}(t) + \frac{B}{mgl}\dot{\theta}(t) + sin(\theta(t)) = \frac{1}{mgl}\tau(t) \tag{39}$$

The normalised input sets, whose universe of discourse are $[-1, 1]$, and the centre of gravity of the output set in $\{-1, -0.5, 0, 0.5, 1\}$ (see Figure 25) is as follows:

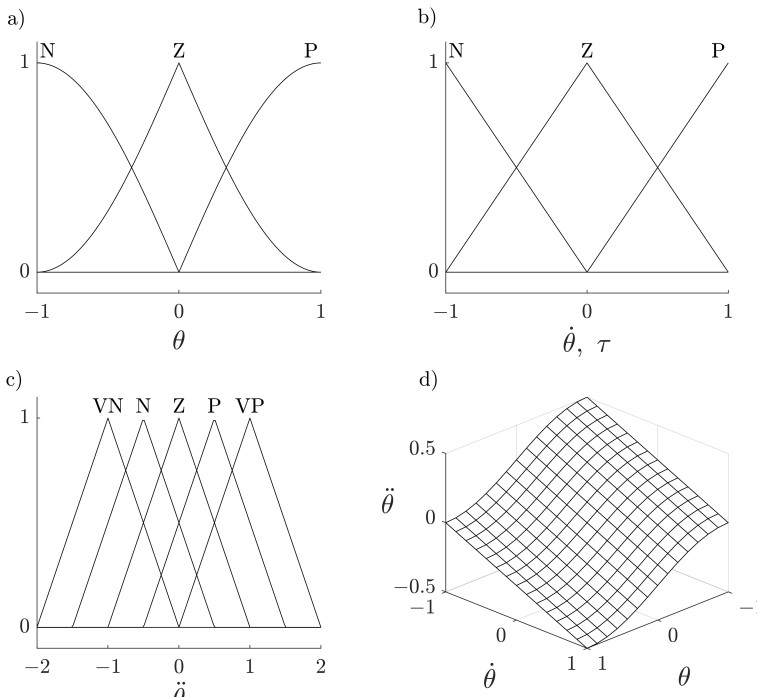

**Figure 25.** Fuzzy sets; (**a**) input set $\theta$, (**b**) input sets $\dot{\theta}$ *and* $\tau$, (**c**) output set $\ddot{\theta}$ centre of gravity and (**d**) output surface between $\ddot{\theta}$, $\dot{\theta}$ and $\theta$.

Figure 26 shows the input and output scaling factors.

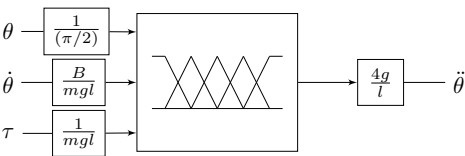

**Figure 26.** Simple pendulum input and output scaling factors.

The resultant Mamdani model is as follows:

$$R^{(1)} : \text{IF } \theta \text{ is } N \text{ AND } \dot\theta \text{ is } N \text{ THEN } \ddot\theta = 4\frac{g}{l}$$

$$R^{(2)} : \text{IF } \theta \text{ is } N \text{ AND } \dot\theta \text{ is } Z \text{ THEN } \ddot\theta = 2\frac{g}{l}$$

$$R^{(3)} : \text{IF } \theta \text{ is } N \text{ AND } \dot\theta \text{ is } P \text{ THEN } \ddot\theta = 0$$

$$R^{(4)} : \text{IF } \theta \text{ is } Z \text{ AND } \dot\theta \text{ is } N \text{ THEN } \ddot\theta = 2\frac{g}{l}$$

$$R^{(5)} : \text{IF } \theta \text{ is } Z \text{ AND } \dot\theta \text{ is } Z \text{ THEN } \ddot\theta = 0$$

$$R^{(6)} : \text{IF } \theta \text{ is } Z \text{ AND } \dot\theta \text{ is } P \text{ THEN } \ddot\theta = -2\frac{g}{l}$$

$$R^{(7)} : \text{IF } \theta \text{ is } P \text{ AND } \dot\theta \text{ is } N \text{ THEN } \ddot\theta = 0$$

$$R^{(8)} : \text{IF } \theta \text{ is } P \text{ AND } \dot\theta \text{ is } Z \text{ THEN } \ddot\theta = -2\frac{g}{l}$$

$$R^{(9)} : \text{IF } \theta \text{ is } P \text{ AND } \dot\theta \text{ is } P \text{ THEN } \ddot\theta = -4\frac{g}{l}$$

$$R^{(10)} : \text{IF } \tau \text{ is } N \text{ THEN } \ddot\theta = -2\frac{g}{l}$$

$$R^{(11)} : \text{IF } \tau \text{ is } Z \text{ THEN } \ddot\theta = 0$$

$$R^{(12)} : \text{IF } \tau \text{ is } P \text{ THEN } \ddot\theta = 2\frac{g}{l}$$

The result of the fuzzy model applied to the simple pendulum can be seen in Figure 27. The graph was elaborated upon using different friction coefficients: B = [0, 0.2, 0.4, 0.6, 0.8, 1], $m = 0.25$ kg, $l = 0.8$ m and $g = 9.81$ m/s$^2$. It was also compared with the dynamic model.

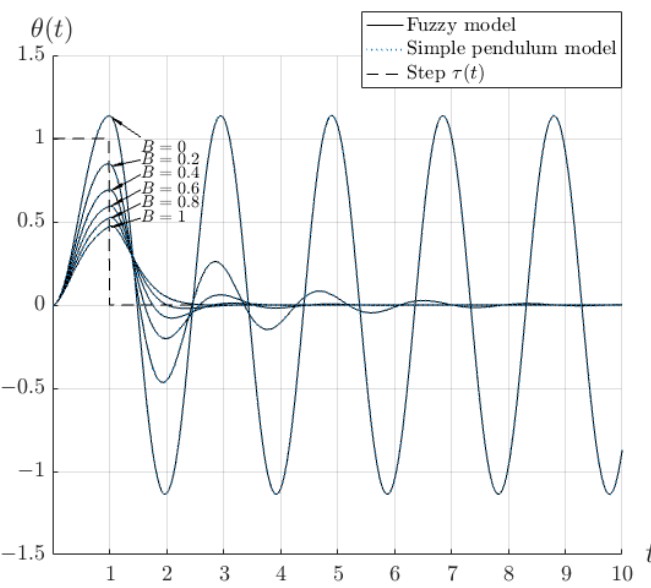

**Figure 27.** Simple pendulum response.

## 6. Conclusions

In this article, a dynamic analysis of first-order and second-order systems was performed. First, a non-linear first-order dynamic system is modelled, and the settling time is determined using the Mamdani fuzzy model. It is deduced that the higher the slope, as the derivative approaches zero, the shorter the settling time, and the lower the slope, the longer the settling time is. Furthermore, it has been shown that the settling time is accurately obtained by switching from a triangular membership function formulation to linguistic variables for non-linear models.

Secondly, the specifications of the time-domain features, such as overshoot percentage and peak time, were graphed and compared with the second-order model features. The percentage of Mp was determined to reach zero when the damping ratio $\zeta$ is equal to or greater than one. The percentage of Mp reaches zero when the damping ratio $\zeta$ is similar to or greater than one. In both cases, this occurs due to the linguistic hedge of concentration in response to a step input and dilation in response to a step-down input. In these cases, one is shorter than the peak time in second-order linear systems.

Furthermore, when the percentage of Mp reaches zero and the damping ratio $\zeta$ is less than one, the linguistic hedge of concentration is in response to a step-down and dilation in response to a step. In the applications section, examples of tank and pendulum models were tested in which the settling time was determined. The performance of different models was compared, and it was demonstrated that the fuzzy model is the best.

**Author Contributions:** Conceptualisation, F.M. and B.M.A.-H.; methodology, F.M and B.M.A.-H.; software, M.B.; validation, M.B., F.M. and B.M.A.-H.; formal analysis, M.B., F.M. and B.M.A.-H.; investigation, M.B., F.M. and B.M.A.-H.; writing—review and editing, M.B., F.M. and B.M.A.-H. All authors have read and agreed to the published version of the manuscript.

**Funding:** This publication is part of the R&D project "Cognitive Personal Assistance for Social Environments (ACOGES)", reference PID2020-113096RB-I00, funded by MCIN/AEI/10.13039/501100011033: "and ESF Investing in your future".

**Institutional Review Board Statement:** Not applicable.

**Informed Consent Statement:** Not applicable.

**Data Availability Statement:** Not applicable.

**Acknowledgments:** **Conflicts of Interest:** The authors declare no conflict of interest.

**Sample Availability:** Samples of the compounds are available from the authors.

## Abbreviations

The following abbreviations are used in this manuscript:

| | |
|---|---|
| T-S | Takagi–Sugeno |
| $t_s$ | settling time |
| Mp | per cent overshoot |
| tp | peak time |
| $t_d$ | delay time |
| NLARX | non-linear ARX |

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
