# Peer review of "Dynamic Analysis of Fuzzy Systems"

_applsci, doi:10.3390/app13031934_

Round 1
Reviewer 1 Report
This is a very nice paper, the length of which is absolutely appropriate. It brings a novelty to the scientists and researchers in the studied field. The ideas are clearly formulated, and the paper is well structured. As far as the research content is concerned, it is interesting and suitable for the readers of this journal. As an expert in the field, I consider the topic of the paper interesting. It fulfills all necessary quality standards. The tables are of a good quality, too. They are readable and the presented results are clear. Figures meet the formal requirements; they do not need to be changed. Authors of the paper provide the readers with plenty of relevant referenced papers. Based on all mentioned characteristics of the paper I recommend accepting and publish it in its current form. I do not feel to be qualified enough to judge the language aspects of the manuscript, but I could understand it quite well. Nice work.
Author Response
Dear Reviewer:
The authors appreciate the reviewer's time and attention to the paper. We thank you for your encouraging comments, it motivates us to continue working on this issue, and we are pleased that you enjoyed the article.
Kind regards,
The Authors.
Manuel Barraza
Fernando Matía
Basil Mohammed Al-Hadithi

Reviewer 2 Report
This work presents a method for analyzing the dynamic behavior of non-linear systems using a fuzzy Mamdani model. The model uses linguistic variables to describe the system and can be used to analyze the time domain characteristics of second-order systems, such as overshoot and peak time. Scaling factors make it easier to adjust the variables and find the system model. The proposed method is demonstrated through case studies involving the analysis of a tank and a simple pendulum.
The work is interesting, but has some serious flaws:
- Introduction is too short and very general. Many articles are cited in groups; a few are very old (older than 2000). The introduction should present the current state of the problem, which is missing. It should present also why this problem is so important.
- The main content of the article is very chaotic. The mathematical parts are correct. All the presented information about the used membership functions should be obvious to the reader, but now they are not. The content in subsections 3 and 4 should be rearranged. What are the main differences between dilation, concentration, and first (second order) models?
- Subsection 5 introduces a simple pendulum after the tank case study. Why introduce these two separate examples? What was it for?
- Subsection 5 introduces NLARX without explanation. What for?
- Article presents a study of the Mamdami models. Why not include Sugeno models?
Author Response
Dear Reviewer:
The authors are grateful for the reviewer's time and comments. The comments are valuable and contribute to improving the work. In this sense, we have modified it to comply with all the reviewer's corrections. In the new manuscript, we have marked the modifications made to the original text in yellow.
We hope that the work we have done will be able to answer the questions and comments raised. If this is not the case, all authors can resolve questions or proceed with further revisions.
"Please see the attachment."
Kind regards,
The Authors.
Manuel Barraza
Fernando Matía
Basil Mohammed Al-Hadithi
Point 1: ”Introduction is too short and very general. Many articles are cited in groups; a few are very old (older than 2000). The introduction should present the current state of the problem, which is missing. It should present also why this problem is so important.”
Response 1: We are grateful for the comment, we fully agree with the reviewer, and for this reason, the introduction was modified, starting with the Mamdani and Takagi-Sugeno Model and justifying why the Mamdani model was used. It explains why the linguistic variables are used and how it helps identify the model. Another important point is that in the literature on fuzzy model identification, only few publications deal with the dynamic properties of the fuzzy model. With this work, we want to contribute to the dynamic properties of the fuzzy model to obtain parameters such as settling time, peak time, and overshoot. (12 current publications were added)
Point 2: “The main content of the article is very chaotic. The mathematical parts are correct. All the presented information about the used membership functions should be obvious to the reader, but now they are not. The content in subsections 3 and 4 should be rearranged. What are the main differences between dilation, concentration, and first (second order) models?”
Response 2: Thank you very much for this comment. The corresponding changes were made to give the correct order to the content. The content of subsections 3 and 4 has been ordered, following the order of the summary and the proposed order indicated in the introduction. For concentration and dilation, examples were added better to understand the linguistic variable and linguistic hedges.
The first-order real one-pole model is used to obtain the dynamic parameter of settling time, and the second-order complex two-pole model for the dynamic parameters of Mp and tp.
Point 3: “Subsection 5 introduces a simple pendulum after the tank case study. Why introduce these two separate examples? What was it for?”
Response 3: Thank you for your comment. We hope this clarifies your questions. Two examples were used, one for the first order system, the model of a tank was obtained using the linguistic variable of dilation in the input h, and the settling time was calculated. The second example of a simple pendulum used the second-order fuzzy model from subsection 4.1. The membership function was modified to fit the model with a normalized sinusoidal function. This was applied to the input, showing that a proper selection of the membership function of the model was equal to the mathematical model. The text was added in subsection 5.2, explaining this difference.
Point 4: “Subsection 5 introduces NLARX without explanation. What for?”
Response 4: Thank you for your question. A detailed explanation was added to clarify the use and why it was compared to this one. The NLARX model is used to model non-linear systems that are influenced by exogenous variables. As the system has a time delay at the input, this time delay can be considered an exogenous variable, as it is a factor that influences the relationship between input and output.
Point 5: “Article presents a study of the Mamdani models. Why not include Sugeno models?”
Response 5: We appreciate this question, and to address this concern, the use of the Mamdani model was justified in the introduction. A brief comparison between Mamdani and T-S was made at the beginning of the article. As explained in the introduction, regardless of the membership function used, the T-S model will not precisely match the empirical model.
